# Siboglinidae Tubes as an Additional Niche for Microbial Communities in the Gulf of Cádiz—A Microscopical Appraisal

**DOI:** 10.3390/microorganisms8030367

**Published:** 2020-03-05

**Authors:** Blanca Rincón-Tomás, Francisco Javier González, Luis Somoza, Kathrin Sauter, Pedro Madureira, Teresa Medialdea, Jens Carlsson, Joachim Reitner, Michael Hoppert

**Affiliations:** 1Institute of Microbiology and Genetics, Georg-August-University Göttingen, 37077 Göttingen, Germany; kathrin.sauter@yahoo.com (K.S.); mhopper@gwdg.de (M.H.); 2Göttingen Centre of Geosciences, Georg-August-University Göttingen, 37077 Göttingen, Germany; jreitne@gwdg.de; 3Marine Geology Dv., Geological Survey of Spain, IGME, 28003 Madrid, Spain; fj.gonzalez@igme.es (F.J.G.); l.somoza@igme.es (L.S.); t.medialdea@igme.es (T.M.); 4Estrutura de Missão para a Extensão da Plataforma Continental (EMEPC), 2770-047 Paço de Arcos, Portugal; pedro.madureira@emepc.mm.gov.pt; 5Area 52 Research Group, School of Biology and Environmental Science/Earth Institute, University College Dublin, Dublin 4, Ireland; jens.carlsson@ucd.ie; 6Göttingen Academy of Sciences and Humanities, 37073 Göttingen, Germany

**Keywords:** Siboglinidae, cold seeps, chemosynthesis, TEM, SEM, biomineralization

## Abstract

Siboglinids were sampled from four mud volcanoes in the Gulf of Cádiz (El Cid MV, Bonjardim MV, Al Gacel MV, and Anastasya MV). These invertebrates are characteristic to cold seeps and are known to host chemosynthetic endosymbionts in a dedicated trophosome organ. However, little is known about their tube as a potential niche for other microorganisms. Analyses by scanning and transmission electron microscopy showed dense biofilms on the tube in Al Gacel MV and Anastasya MV specimens by prokaryotic cells. Methanotrophic bacteria were the most abundant forming these biofilms as further supported by 16S rRNA sequence analysis. Furthermore, elemental analyses with electron microscopy and energy-dispersive X-ray spectroscopy point to the mineralization and silicification of the tube, most likely induced by the microbial metabolisms. Bacterial and archaeal 16S rRNA sequence libraries revealed abundant microorganisms related to these siboglinid specimens and certain variations in microbial communities among samples. Thus, the tube remarkably increases the microbial biomass related to the worms and provides an additional microbial niche in deep-sea ecosystems.

## 1. Introduction

Chemosynthetic fauna is widely distributed and often found in deep-sea areas of active fluid seepage where oxygen levels are normally low, such as in hydrothermal vents and cold seeps. Yet, they can also be found in other reduced environments, such as whale and wood falls [1,2,3,4,5]. While the composition of the seepage fluids is variable, some (micro)organisms have adapted to use some of the most abundant constituents as their energy and/or carbon source, i.e., methane and sulfur compounds. These chemosynthetic organisms sustain these ecosystems by acting as primary producers and supplying the higher trophic levels with nutrients [6,7]. They also provide the hard substrate that most filter-feeders need to settle and develop. As an example, authigenic carbonates formed due to the anaerobic oxidation of methane (AOM) [8] act as optimal substrate for sponges and corals.

Characteristic fauna found in these ecosystems include bivalves (within the Mytilidae, Vesicomyidae, Solemyidae, Thyasiridae and Lucinidae families) [4,9,10,11], annelids (within the Alvinellidae and Siboglinidae families) [3,11,12], and protozoans such as ciliates [13,14], that live in symbiosis with these chemolitoautotrophic bacteria. These bacteria provide their hosts with a rich source of nutrients in a high methane and sulfur environment where they are protected inside the hosts.

Tube fossils of siboglinid worms from vent sites are dated from the Silurian period, ca. 430 Ma ago [15,16,17]. Taxonomic groups of the Siboglinidae family are described as a fundamental part of the core chemosynthetic community in reduced environments [16]. Siboglinids are normally found in the oxic/anoxic interface, as the symbiotic microorganisms require oxygen as the electron acceptor to oxidize methane or sulfide. Symbionts are normally found in the anterior part of the siboglinid inside the chitin tube [18], in contact with the water column, from where they acquire the oxygen, while the posterior part is inside the reduced sediment, from where they collect the nutrients for their endosymbionts [2].

Adult siboglinids lack gut and rely on their endosymbiotic bacteria for nutrition, which are located in bacteriocytes inside the highly vascularized trophosome organ [12,19,20]. Thiotrophic Gammaproteobacteria are the most common microorganisms found in siboglinid trophosomes [21]. However, methanotrophic symbionts in siboglinid species from methane vents have also been reported, i.e., *Siboglinum poseidoni* [22,23] and *Sclerolinum contortum* [24]. To date, all methanotrophic symbionts identified are related to type I methanotrophs from the Gammaproteobacteria, while type II methanotrophs from the Alphaproteobacteria have not been found as symbionts in any marine invertebrate [21].

While most studies are focused on the interaction between siboglinids and their endosymbionts, few studies have reported the presence of microorganisms colonizing the tube or considered these tubes as potential niche for other chemosynthetic and non-chemosynthetic microorganisms. Microbial communities have not only been described on the outside of the tubes of *Riftia pachyptila* [25], *Lamellibrachia* sp. [26] and *Escarpia* sp. [27], but they have also been found in the internal face of the tube [26]. Furthermore, in 2015, Georgieva et al. found bacterial biofilms inside the tube of *Alvinella* sp. worms (Alvinellidae family), acting as one more concentric layer of the multiple layers that constitute the tube of the worms. These extraneous microbial inner cores were proposed to be formed due to the colonization of the surface of the tube followed by its normal progressive mineralization [17].

Tubeworms live in dense colonies and their tubes provide a considerable increase in solid, microbially colonizable surface. This implies an increase in the microbiota of the worms as well as in the total microbial biomass of cold seep ecosystems. Furthermore, the microbial community determines the composition of the different layers of the tube. Thus, we examined different specimens of small siboglinids recovered from four mud volcanoes in the Gulf of Cádiz, i.e., El Cid MV, Bonjardim MV, Al Gacel MV, and Anastasya MV [28,29,30]. We used light microscopy, transmission electron microscopy (TEM) and scanning electron microscopy coupled to EDX (SEM-EDX) for the characterization of the tube and tissue of these specimens. Illumina next generation amplicon sequencing of 16S rRNA genes of prokaryotes present on the samples was used to support preliminary characterization of the siboglinids’ microbiota by imaging approaches.

## 2. Materials and Methods

### 2.1. Specimen Collection

Field experiments were approved by the Spanish Ministry of Science, Innovation and Universities (project SUBVENT CGL2012-39524-C02 and project EXPLOSEA CTM2016-75947) and the Irish Marine Institute (project Deep-Links: Ecosystem services of deep-sea biotopes CE15012). Different small siboglinid specimens were recovered from different mud volcanoes (MV) in the Gulf of Cádiz (Table 1). El Cid MV, Bonjardim MV and Al Gacel MV were sampled during the 2014 Subvent-2 cruise (R/V Sarmiento de Gamboa) using the Portuguese remote operated underwater vehicle (ROV) Luso, while the Anastasya MV was sampled during the 2015 Deep-Links cruise (R/V Celtic Explorer) using the ROV Holland (Figure 1). Both underwater vehicles carry a CTD sensor (ROV Luso also has as CH_4_ sensor) to measure different water variables such as depth, temperature, oxygen, pH, and redox potential. From each mud volcano between 5 and 10 specimens were fixed for transmission and scanning electron microscopy (TEM and SEM, respectively), and between 10 and 15 specimens were stored in ethanol or kept at −80 °C for staining technics and DNA analysis.

### 2.2. Transmission Electron Microscopy (TEM)

Specimens from Al Galcel MV and Anastasya MV were fixed in 2.5% (w/v) glutaraldehyde. After washing several times with phosphate-buffered saline (137 mM NaCl, 2.7 mM KCl, 10 mM Na_2_HPO_4_, 1.8 mM KH_2_PO_4_, pH 7.4), a dehydration series was performed (15%, 30%, 50%, 70%, 95%, and 100% aqueous ethanol solution), followed by embedding the samples with Medium LR white resin (Plano, Wetzlar, Germany). Polymerization of the resin was at 60 °C during 24 h. A milling tool (TM 60, Fa. Reichert & Jung, Vienna, Austria) was used to make a truncated pyramid on the gelatin capsules. Furthermore, an ultramicrotome (Ultracut E, Reichert & Jung, Vienna, Austria) and glass knives were used for obtaining ultrathin sections of the sample. Ultrathin sections were 80 nm in thickness, mounted on 300 mesh specimen Grids (Plano, Wetzlar, Germany), further stained with 4% (w/v) uranyl acetate (positive stain). The sections were inspected in a Jeol EM 1011 transmission electron microscope (Jeol, Eching, Germany).

### 2.3. Scanning Electron Microscopy (SEM) and Energy-Dispersive X-Ray Spectroscopy (EDX) Analysis

While specimens from El Cid MV and Anastasya MV were dehydrated without prior fixation, specimens from Al Gacel MV were fixed in 2.5 % (w/v) glutaraldehyde to maintain the native organic structures. After washing several times with phosphate-buffered saline, a dehydration series was performed (15%, 30%, 50%, 70, 80%, 90%, 95%, and 100% aqueous ethanol solution), followed by hexamethyldisilazane (HMDS; Sigma-Aldrich, Germany) in order to avoid drying artifacts. Samples were mounted on SEM sample holders and sputtered with Au–Pd (13.9 nm for 120 s). They were further visualized in a SEM LEO 1530 Gemini (Zeiss, Oberkochen, Germany) combined with an INCA X-ACT EDX.

### 2.4. Fluorescent Staining of Chitin Tubes

Tubes of specimens recovered from the Al Gacel MV were stained with calcofluor white (Merck, Darmstadt, Germany) to identify the chitin tube. Following previous staining of the samples, they were fixed on a slide and embedded in paraffin followed by a graded ethanol series (100%, 90%, 70%, and 50%). Afterwards, one drop of staining and one drop of KOH 10% were placed onto the slide with the sample. The samples were examined under normal light and a UV filter with an excitation ranges between 450 and 490 nm of a Zeiss Axioplan microscope (Zeiss, Oberkochen, Germany).

### 2.5. DNA Extraction and Amplification of Bacterial and Archaeal 16S rRNA Genes

Between 10 and 15 specimens (bulk of empty tubes and worms inside of tubes) from El Cid MV, Bonjardim MV, Al Gacel MV and Anastasya MV were used for this analysis. Total DNA was isolated with Power Soil DNA Extraction Kit (MO BIO Laboratories, Carlsbad, CA, USA) according to manufacturer’s instructions. Bacterial amplicons of the V3 – V4 region were generated with the primer set S-D-Bact-0341-b-S-17 / S-D-Bact-0785-a-A-21, with added Illumina adapter overhang nucleotide sequences (forward primer: 5′-TCG TCG GCA GCG TCA GAT GTG TAT AAG AGA CAG CCT ACG GGN GGC WGC AG-3′; reverse primer: (5′-GTC TCG TGG GCT CGG AGA TGT GTA TAA GAG ACA GGA CTA CHV GGG TAT CTA ATC C-3′) [31]. Likewise, archaeal amplicons of the V3 – V4 region were generated with the forward primer based on Arch514Fa (5′-TCG TCG GCA GCG TCA GAT GTG TAT AAG AGA CAG GGT GBC AGC CGC CGC GGT AA-3′) and the reverse primer (5′-GTC TCG TGG GCT CGG AGA TGT GTA TAA GAG ACA GCC CGC CAA TTY CTT TAA G-3′) [32]. The PCR reaction mixture for bacterial DNA amplification, with a total volume of 50 µl, contained 1 U Phusion high fidelity DNA polymerase (Biozym Scientific, Oldendorf, Germany), 5% DMSO, 0.2 mM of each primer, 200 μM dNTP, 0.15 μL of 25 mM MgCl_2_, and 25 ng of isolated DNA. Furthermore, PCR protocol for bacterial DNA amplification was: initial denaturation for 1 min at 98 °C, 25 cycles of 45 s at 98 °C, 45 s at 60 °C, and 30 s at 72 °C, and a final extension at 72 °C for 5 min.

The PCR reaction mixture for archaeal DNA amplification was similarly prepared but contained 1 μL of 25 mM MgCl_2_ and 50 ng of isolated DNA. PCR protocol for archaeal DNA amplification was: initial denaturation for 1 min at 98 °C, 10 cycles of 45 s at 98 °C, 45 s at 63 °C, and 30 s at 72 °C, 15 cycles of 45 s at 98 °C, 45 s at 53 °C, and 30 s at 72 °C, and a final extension at 72 °C for 5 min. PCR products were purified using the GeneRead Size Selection Kit (QIAGEN GmbH, Hilden, Germany).

### 2.6. Bioinformatic Processing of Amplicons

300 Paired-end (300PE) sequencing of the amplicons was performed in the Göttingen Genomics Laboratory (Göttingen, Germany). Paired-end sequences were merged, and sequences containing unresolved bases and reads shorter than 305 base pairs (bp) were removed using PANDAseq v2.11 [33], employing the PEAR algorithm v0.9.8 [34]. Non-clipped forward and reverse primer sequences were removed by employing cutadapt v1.15 [35]. QIIME 1.9.1 was used to process the amplicon sequences [36]. The sequences were dereplicated and checked for chimeric sequences (de novo). Sequences were clustered at 97% sequence identity to operational taxonomic units (OTUs). The taxonomic classification of the OTU sequences was performed against the SILVA database 132 employing the assignment method implemented in Mothur [37]. Extrinsic domain OTUs, chloroplasts, and unclassified OTUs were removed from the dataset. Sample comparisons were performed at the same surveying effort, utilizing the lowest number of sequences by random subsampling (30,563 reads for bacteria, 4080 reads for archaea). Beta-diversity was calculated using Unifrac statistics [38] to determine the distances between samples. Principal coordinates analysis (PCoA) plots representing the data was visualized using EMPeror tool [39]. R programming was used to construct heatmaps representing the relative abundances of bacterial and archaeal communities in each sample. The paired-end reads of the 16S rRNA gene sequencing were deposited in the National Center for Biotechnology Information (NCBI) in the Sequence Read Archive SRR8944123 with the accession number PRJNA533037.

## 3. Results

### 3.1. Samples and in Situ Variables’ Measurement

Siboglinid specimens were recovered from different mud volcanoes at sites where reduced sediment was observed. Exact location of the samples, as well as data collected from the ROVs’ sensors (CH_4_ and CTD) and are shown in Table 1. El Cid MV and Bonjardim MV specimens were sampled from grey mounds (Figure 1B,C). The El Cid MV siboglinid-sample was collected from the first 5 cm of a sediment push-core, while the Bonjardim MV sample, which was found in a mud breccia with a strong hydrogen sulfide smell, was collected using a suction sampler. Siboglinid specimens recovered from the Al Gacel MV were located in a pockmark, beneath an AOM-derived carbonate and facing an active bubbling seepage (Figure 1D) [40]. Furthermore, Anastasya MV specimens were obtained from a field of *Beggiatoa*-like biofilms (Figure 1E). All specimens were about 100 µm width and not more than 15 cm in length. Their tubes had a light-brownish color. No intensive morphological identification could be made; however, based on their size and external appearance they are likely *Siboglinum* sp. or *Sclerolinum* sp. specimens.

### 3.2. Imaging of endosymbionts

During transmission and scanning electron microscopy, worm tissues were only observed in the Al Gacel MV (see supplementary data Appendix A) and Anastasya MV samples (Figure 2). The other samples consisted of empty tubes. SEM micrographs from one specimen of Anastasya MV revealed the posterior region of the worm (Figure 2A) with a segmented opisthosoma and the trophosome (Figure 2B). A hole in the trophosome exposed abundant bacteria inside (Figure 2C,D). These bacteria were cocci of ca. 0.5 µm in diameter (Figure 2D).

### 3.3. Structure and Composition of the Tubes

The fluorescent stain calcofluor white is an indicator for polysaccharides such as chitin, which is part of the organic matrix of siboglinids’ tubes. Sections of empty tubes from the Al Gacel MV expressed high fluorescence when observed under UV-light with a 09 Zeiss filter, with excitation wavelength ranges between 450 and 490 nm (Figure 3). Furthermore, an external microbial layer de-attached from the tube (most likely due to handling of the sample) was slightly fluorescent (Figure 3B).

SEM micrographs revealed transversal-segmented tubes, which were covered by minerals (from El Cid MV, Figure 4A), a thick biofilm (from Al Gacel MV, Figure 4B) or putative remains of microbial extracellular polymeric substances or EPS (from Anastasya MV, Figure 4C). Disrupted tubes revealed their composition of multiple concentric layers between 6 and 10 µm in thickness (Figure 4D). Some of the layers displayed a filamentous matrix, with attached globular particles of ca. 200 nm in diameter (Figure 4E). Layers consisting of these particles show a significant silica signal in EDX analysis (see supplementary data Appendix A). Other layers contained significant amounts of iron, sulfur and calcium, without notable differentiation between them. Detailed interpretation of EDX-analysis is discussed in supplementary data. Furthermore, microbial cells were observed in the internal surface of the tube from Al Gacel MV (Figure 4D and Figure 5G,H). A model of the different layers observed in a tube is shown in Figure 4F.

### 3.4. Microbial Biofilm of the Tubes

TEM and SEM micrographs from the Al Gacel MV revealed a high microbial colonization of the outside surface of the tube (Figure 4B and Figure 5). The biofilm was up to 10 and 20 µm thick (Figure 5A). Bacteria with intracytoplasmic membranes arranged as known for methanotrophic proteobacteria were the most abundant along the tube, forming densely packed cell-aggregations (Figure 5A–C). Other microbial morphotypes were observed, i.e., prosthecate, rod shaped, helically shaped, and filamentous microorganisms (Figure 5D–F). Rod-shaped microorganisms were also observed attached to the inside surface of the tube (Figure 5G,H) Furthermore, some microorganisms appeared to be actively penetrating the chitin tube (Figure 5I). Similarly, siboglinids’ tubes from Anastasya MV under the TEM revealed a biofilm on the external tube face. However, the biofilm appeared to be in a degradation process, because cells appeared as “ghosts” (only cell walls, no cytosolic contents were visible; Figure 6). Remains of EPS forming similar cell-aggregations to the ones observed in Al Gacel MV tube indicate abundance of methanotrophic bacteria (Figure 6A). Embedded remains of microorganisms inside the tube were also observed (Figure 6B).

### 3.5. Prokaryotic Community Composition

Bacterial and archaeal 16S rRNA gene libraries revealed relative abundances of taxa typically found thriving in the water column, such as Acidobacteria, Actinobacteria, Bateriodetes, Chloroflexi, Thermoplasmata, Woesearchaeota, and *Candidatus* Nitrosopumilus (Figure 7) [41,42].

Sulfide-oxidizing bacteria are detected in all samples, with a high representation in the El Cid MV sample, being mostly *Thiohalophilus* and bacteria from the Thiotrichaceae family (Figure 7A). *Sedimenticola* endosymbionts, which are also sulfide-oxidizing bacteria, are abundant in Al Gacel MV specimens, as well as Desulfobacterales sulfate-reducers. In fact, sulfate reducers are highly abundant (>15%) in Al Gacel MV and Anastasya MV samples, while in El Cid MV and Bonjardim MV they represent 3% of the total relative abundance (Figure 7A). In Anastasya MV sample, Marine Methylotrophic group 2 (MMG-2) methanotrophic bacteria, and *Desulfobacter* sulfate-reducing bacteria are highly abundant (Figure 7A). Additionally, *Methylotenera* methylotrophic bacteria taxa are also representative in Al Gacel MV (7%). In Al Gacel MV and Anastasya MV up to 50 % of the bacteria are represented by methane-oxidizing, sulfide-oxidizing and sulfate-reducing bacteria (Figure 7C). Likewise, *Chitinivibrionia* (known chitin degraders) were detected in all our samples, especially in Anastasya MV (Figure 7A).

The archaeal community profile was dominated by Woeserarchaeota (or DHVEG-6, Nanoarchaeota), followed by methane-oxidizing archaea (ANME-1 and ANME-2) as the second most abundant taxa, except in Anastasya MV where methanogens are slightly more abundant (Figure 7B,D). Additionally, methanogenic archaea were homogeneous among the samples, except in the Al Gacel MV where they were almost absent (Figure 7B).

Beta-diversity among the samples showed substantial differences in the microbial communities (Figure 7C,D). The first and the second principal coordinates (PCoA1 and PCoA 2, respectively) revealed short distances between El Cid MV and Bonjardim MV bacterial communities (Figure 7C), and between El Cid MV and Anastasya MV archaeal communities (Figure 7D).

## 4. Discussion

### 4.1. Endosymbionts in Siboglinidae Worms

Since siboglinids were discovered in the 1900s and described by Caullery in 1914 [43], researchers have collected data on their life history characteristics and, in particular, adaptations allowing them to survive in reduced environments at high hydrogen sulfide concentrations and low oxygen [12]. To date, it has been established that these tube-dwelling annelids harbor chemolithoautotrophic endosymbionts in the super-vasculated trophosome [19,20]. Those endosymbionts are facultative free-living bacteria which are acquired from the environment by the worms during their juvenile stage, when their guts are reduced [44,45]. Once they become adults, they have established a permanent mutualistic microbe-animal symbiosis, with the host lacking gut and acquiring organic carbon solely from their endosymbionts. This mechanism of obtaining endosymbionts horizontally from the environment has been also described in other animals [46].

Siboglinidae worms mostly harbor thiotrophic bacteria in their trophosomes [16], and only some punctual specimens have been reported to harbor methanotrophic endosymbionts instead, i.e., *Siboglinum poseidoni* recovered from central Skagerrak [22], and *Sclerolinum contortum* sampled at the Haakon Mosby MV [24] and the Gulf of Cádiz [23]. In the current study, endosymbionts from specimens collected in El Cid MV, Al Gacel MV and Anastasya MV were identified. Bacterial 16S rRNA genes from El Cid MV sample presented an OTU with 99 % similarity to a thiotrophic endosymbiont of *Siboglinum* worms recovered from Gemini MV in the Gulf of Cádiz (OTU_0) [47]. Likewise, Al Gacel MV worms revealed high abundance of an OTU with 98% similarity to *Sedimenticola* sp., a thiotrophic endosymbiont of *Sclerolinum contortum* (OTU_4) [48]. Furthermore, we observed bacteria inside of the trophosome (Figure 2C,D) of a small siboglinid from the Anastasya MV (attempted to be classified as *Siboglinum* sp., due to its lack of girdles between the trophosome and opisthosoma; Figure 2A) [20]. Previous studies have also reported *Siboglinum* sp. worms in this volcano [49]. Bacterial 16S rRNA genes revealed that the most abundant methane-oxidizing bacteria in Anastasya MV specimens were related to Marine Methylotrophic Group 2 (MMG-2, Methylococcales; Figure 7A). MMG-2 bacteria have not previously been described as endosymbionts, but MMG-1 and MMG-3 [50]. The high presence of those sequences could indicate that these methanotrophs are actually acting as endosymbionts, as previous studies in Captain Arutyunov MV (Gulf of Cádiz) have reported the presence of these worms living in symbiosis with methanotrophic bacteria [23]. However, those sequences could also be related to the microbial remains observed onto the Anastasya MV tubes (Figure 6) and therefore further studies discriminating between the tube and the tissue of siboglinids worms will be necessary.

### 4.2. The Tube as a New Microbial Niche

External and internal microbial colonization of siboglinid tubes has previously been described [25,26,27]. Light microscopy, SEM and TEM micrographs showed highly colonized tubes in Al Gacel MV specimens (Figure 3, Figure 4B and Figure 5). While microorganisms observed in the internal face of the tube seemed to be a thin microbial layer with isolated microorganisms (Figure 5H), externally, a thick microbial biofilm was composed of mostly methanotrophic bacteria forming cell aggregations (Figure 5A–C), but also filamentous (Figure 5D,E), prosthecate- and spirillum-shaped (Figure 5F), and rod-shaped microorganisms were observed (Figure 5G,H). Microbial 16S rRNA genes from the Al Gacel MV sample revealed the high abundance of bacteria related to Methylococcales (mostly MMG-2), possibly forming the characteristic microbial biofilm. Few sequence-reads were related to Hyphomonodaceae and prosthecate bacteria, which could explain the morphotypes observed on the external biofilm of Al Gacel MV worm (Figure 5F). Rod-shaped bacteria could not be related to any microbial taxa since it is a highly common morphotype (Figure 7A).

Al Gacel MV specimens represent an example of how the tubes of siboglinids provide a viable niche for microorganisms. In fact, microbial biofilms are known to be ecosystems themselves, capable of self-regulation in which all microorganisms are linked and provide each other with stable sources of nutrients and protection [51]. Those microorganisms increment the impact of siboglinid worms in the ecosystem, since they constitute part of the worms’ microbiota and previous studies have pointed out the difference in the microbial composition of siboglinid tubes and the surrounding environment [27]. Consequently, the tube of siboglinids should be considered as an important niche, which increases the microbial biomass and provides a large source of microorganisms, which are part of the microbiota of the worms and ultimately the worm’s holobiont [52].

### 4.3. Tube-Microbe Interaction

The tubes of all siboglinids all have in common that they produce a chitin matrix (Figure 3) secreted by the worm that is incorporated in the tube [18]. Since they are in contact with water and reduced sediments, tubes are rich in minerals and other inorganic compounds, which may vary depending on the environment [53]. High amounts of iron, calcium and sulfur compounds were detected in all the tubes with EDX analysis (supplementary Appendix A), indicating the precipitation of minerals such as pyrite (or ferrous sulfide), aragonite [17,54,55,56], or even iron-silicates [57]. The presence of pyrite and aragonite in reduced environments is usually due to the anaerobic oxidation of methane (AOM) [8]. Microorganisms involved in AOM (sulfate-reducing bacteria and ANME archaea), as well as sulfur oxidizers (which oxidize pyrite), have been found at all sites (Figure 7). Biomineralization of the tube due to the different microbial metabolisms leads to the embedding of such microorganisms inside of the tube (Figure 6B), by coating the external face of the tubes with precipitated minerals. Furthermore, microbial mineralization is accompanied by the precipitation of silica, which interacts with the iron forming iron-silicates and fills up the spaces between the chitin matrix (Figure 4E), ultimately replacing the organic matter [17]. Consequently, composition of siboglinid tubes is principally characterized by microbial layers, chitin-silica layers, precipitated-minerals layers—replacing previous microbial biofilms—and external microbial biofilms. Thus, a model has been proposed showing the composition of the different layers given in the tube of a siboglinid worm based on our results and other studies (Figure 4F) [17,54,55,56].

Additionally, *Chitinovibrionia* chitin-degraders were detected mainly in Anastasya MV (Figure 7A) and Al Gacel MV samples (Figure 5I). This active participation of the biofilm on the tube’s decay could indicate a switch in the microbial community, from a chemosynthetic-based microbial community—which coats and protects the tube—to a heterotrophic-based microbial community. Decay of the chemosynthetic-based biofilm was observed in the Anastasya MV tubes (Figure 4C and Figure 6). Since active emission or hydrocarbon-rich fluids were detected at both sites (see Section 4.4), the decay of these certain tubes is not well understood and could be related to the life-cycle of the worm and not only dependent on the availability of reduced compounds. Additionally, handling of the sample during fixation could have led to the removal of microbial cells and only the EPS extracellular matrix has remained. Further studies focused on the life cycle of these biofilms, as well as their interaction with the worms and impact in the environment, are warranted.

### 4.4. Insights into Siboglinids’ Microbiota

Siboglinidae worms do not only harbor microorganisms in their trophosome, but also on their tubes. Besides, rod-shaped bacteria were observed on an Anastasya MV worm, as potential epibionts (supplementary Appendix A). All these microorganisms associated with Siboglinidae specimens conform the microbiota (or microflora) of these invertebrates. This microbiota is part of its host, and the metabolisms driven by these microorganisms contribute to the total ecological impact of the worm on the environment. Worm and microbiota constitute therefore a unique ecological unit, sometimes referred to as holobiont [52]. Thus, in the same way the community of a mud volcano switches between chemosynthetic and non-chemosynthetic organisms depending on changes of the source of nutrients (i.e., seeped fluids versus organics from photic zone), we observed disparity in the total microbiota of siboglinids sampled from different mud volcanoes and sites with different seepage activity (Figure 7).

El Cid MV and Bonjardim MV specimens were recovered from sites where non-active emission of fluids was detected, and methane concentration was relatively low (70–90 nM and 50–65 nM, respectively; Sánchez-Guillamón et al., 2015) (Figure 8) [58]. The site of El Cid MV from where worms were sampled, was surrounded by non-chemosynthetic fauna (shrimps, fish; Figure 8), while Bonjardim MV sampling was performed in an area where patches of reduced sediment (biofilm-like) and dead bivalves were observed (Figure 8). The sampled sediment with siboglinids from Bonjardim MV emanated a strong smell of hydrogen sulfide, potentially indicating the occurrence of anaerobic oxidation of methane (AOM) in the past. Likewise, the high relative abundance of sulfide oxidizers in El Cid MV samples may also indicate past AOM events (Figure 7C). In fact, DNA related to ANME in both inactive sites was detected (El Cid MV and Bonjardim MV; Figure 7B). Furthermore, sulfate-reducing bacteria are much less abundant in these samples when compared to known active sites (i.e., Al Gacel MV and Anastasya MV; Figure 7A).

On the other hand, the Al Gacel MV and Anastasya MV sampling sites showed bubbling of gas methane hydrates (Figure 8) with methane concentrations as high as 191 nM at the time of *Sclerolinum* worm sampling [58]. At both sites a thick biofilm covering the tube of mainly methanotrophic bacteria was detected—in Anastasya MV specimens only remains of the biofilm were observed—and environmental 16S rRNA genes revealed a higher presence of methane-oxidizing and sulfate-reducing microorganisms in these samples (Figure 7A,B). Since siboglinids normally colonize the oxic-anoxic interface in sites of fluids emission, they optimize at the same time the access to the seeped fluids for both aerobic methane-oxidizing bacteria and anaerobic sulfate-reducing bacteria, allowing them to co-exist in the same niche (the worm).

Interestingly, non-active (El Cid MV, Bonjardim MV) and active (Al Gacel MV, Anastasya MV) sites differ in the composition of their microbial communities (Figure 7C,D). Therefore, our preliminary analyses indicate that seepage activity at these sites directly influence the composition of the microbial community (Figure 7). Since DNA analysis in this study have been used to support imaging approaches and have preliminary results of siboglinids’ total microbiota—without differentiating between tube’s microbiota and worm-tissue’s microbiota—further studies in that direction need to be performed including variation in the microbiota depending on the presence or absence of the living worms.

## 5. Conclusions

Small Siboglinidae specimens appeared to have a higher microbial biomass related to them than previously realized. In addition to the chemosynthetic endosymbionts harbored inside their trophosome, specimens from Al Gacel MV and Anastasya MV revealed that the tube was colonized by a thick microbial biofilm. This external biofilm of the tubes was mostly composed of cell-aggregations of methanotrophic bacteria, but other morphotypes such as filamentous, prosthecate, spirillum-like and rod-shaped bacteria were also observed. Yet, this chemosynthetic-based biofilm seems to contribute to the biomineralization and silification of the tube, conditioning the different concentric layer given in siboglinid tubes. The microbial community on the tube can also participate in the degradation of the chitin tube by chitin-degrading bacteria. Additionally, a preliminary comparison of environmental 16S rRNA gene libraries showed different microbial communities among samples.

## Figures and Tables

**Figure 1 microorganisms-08-00367-f001:**
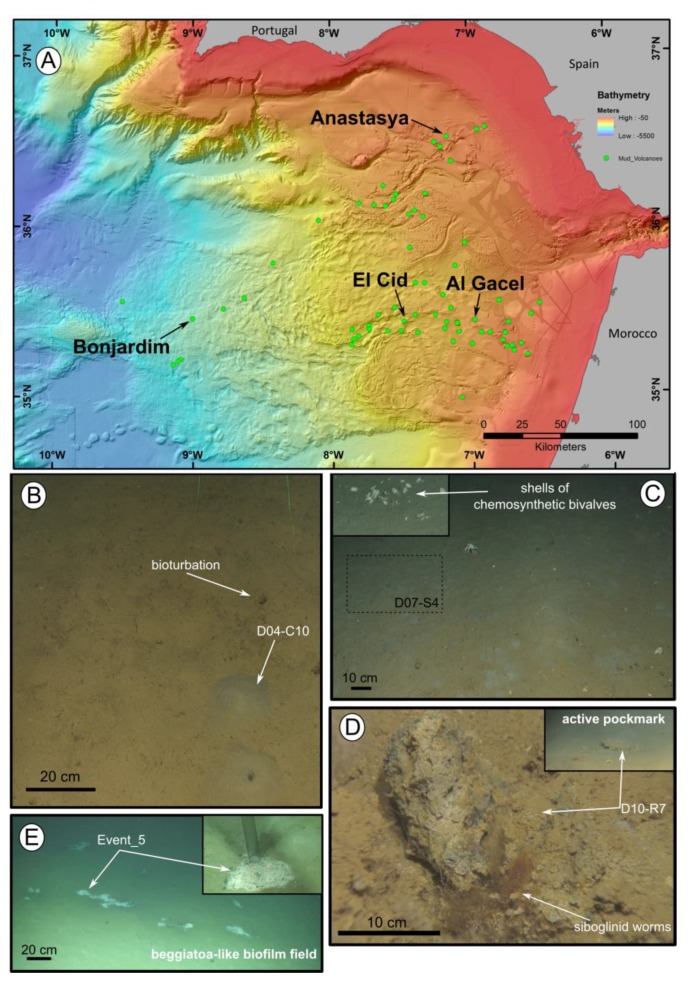
Location of the mud volcanoes sampled for this study in the Gulf of Cádiz and an overview of the sites where samples were recovered. (**A**) General view of the Gulf of Cádiz. Arrows point to mud volcanoes from where the samples were taken. (**B**–**E**) ROV still frames from the different sampling sites. (**B**) El Cid MV. (**C**) Bonjardim MV. (**D**) Al Gacel MV. (**E**) Anastasya MV. Coordinates in Table 1.

**Figure 2 microorganisms-08-00367-f002:**
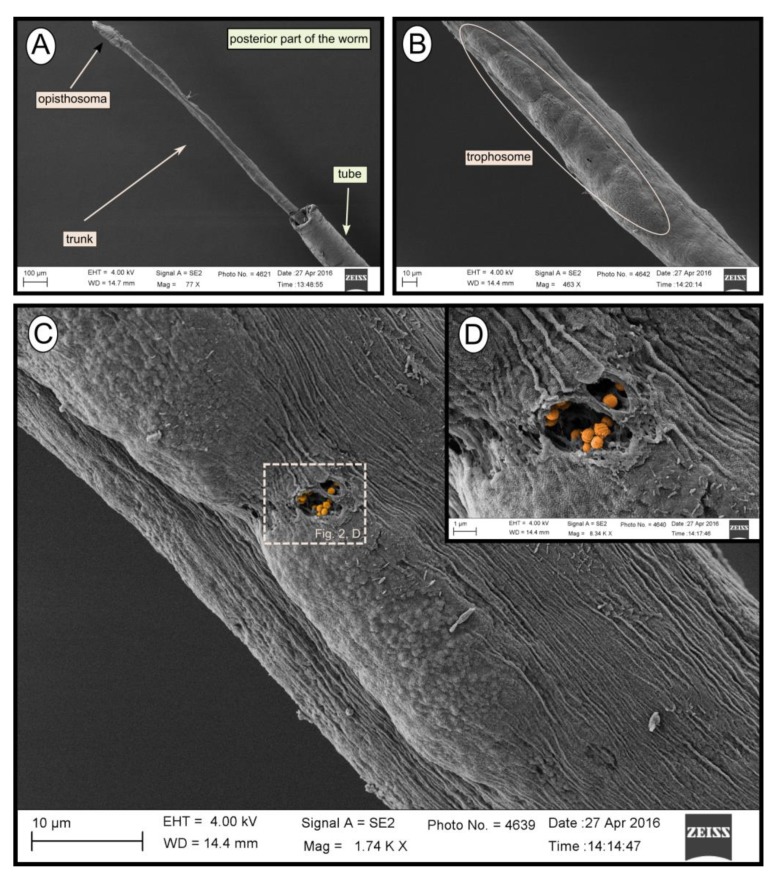
SEM micrographs of one specimen from Anastasya MV. (**A**) General view of the sample. Posterior part of the worm and the tube are visible. (**B**) Trophosome. (**C**,**D**) Closer view to a hole in the trophosome with endosymbiotic bacteria exposed (colored).

**Figure 3 microorganisms-08-00367-f003:**
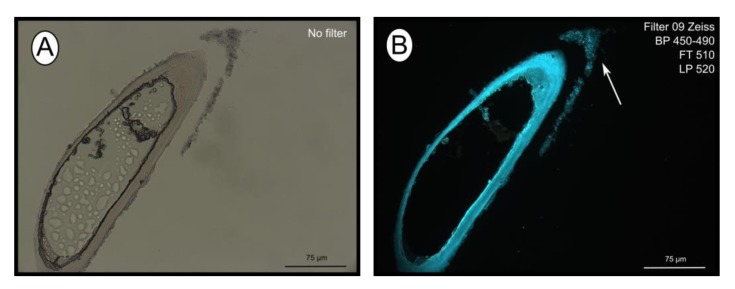
Calcofluor white staining of empty tubes recovered from Al Gacel MV. The fluorescence of the tube indicates the presence of chitin. Same section under normal light (**A**) and using Filter 09 with an excitation range between 450 and 490 nm (**B**). Notice the fluorescence of the tube and of the detached microbial layer (marked with an arrow).

**Figure 4 microorganisms-08-00367-f004:**
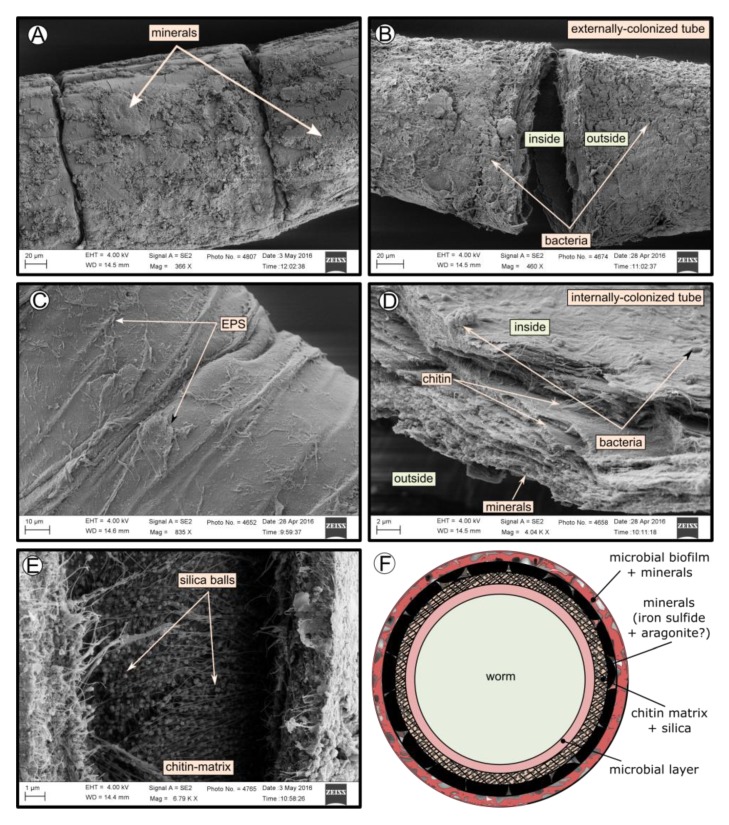
SEM micrographs of the tube of different specimens from different mud volcanoes and the expected arrangement of their layers. (**A**) El Cid MV specimen, with minerals on its external surface. (**B**) Al Gacel MV specimen, with a thick biofilm on its external surface. Microbial colonizers are detailed in Figure 5. (**C**) Anastasya MV specimen with remains of EPS on its external surface. (**D**) Al Gacel MV specimen with bacteria on its internal surface. A multilayer organization of the tube can be observed, chitin layers and minerals can be differentiated. (**E**) Internal layer of chitin with rounded silica from El Cid MV specimen. (**F**) Model of what is expected to be the arranging of the tube.

**Figure 5 microorganisms-08-00367-f005:**
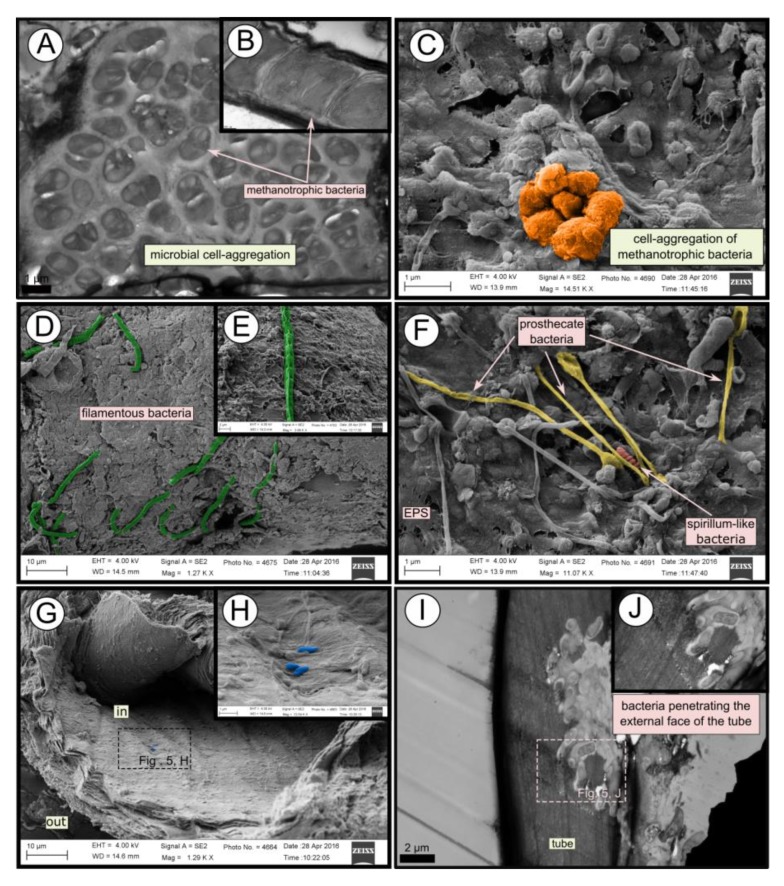
SEM and TEM micrographs of colonized tubes from Al Gacel MV. (**A**–**C**) Methanotrophic bacteria organized in cell-aggregations and expressing intracytoplasmatic membranes. (**D**–**F**) Different microbial morphotypes observed in the biofilm. (**G**,**H**) Rod-shaped microorganisms colonizing the internal surface of the tube. (**I**,**J**) Microbial biofilm penetrating the tube.

**Figure 6 microorganisms-08-00367-f006:**
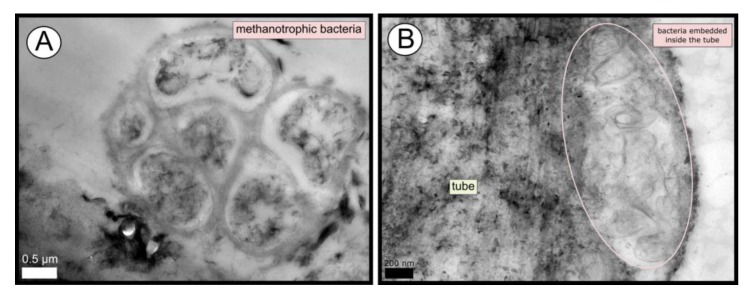
TEM micrographs of remains of a microbial biofilm from tubes of Anastasya MV worms. (**A**) Remains of methanotrophic bacteria are commonly observed. (**B**) Many microorganisms appear to be embedded inside the tube.

**Figure 7 microorganisms-08-00367-f007:**
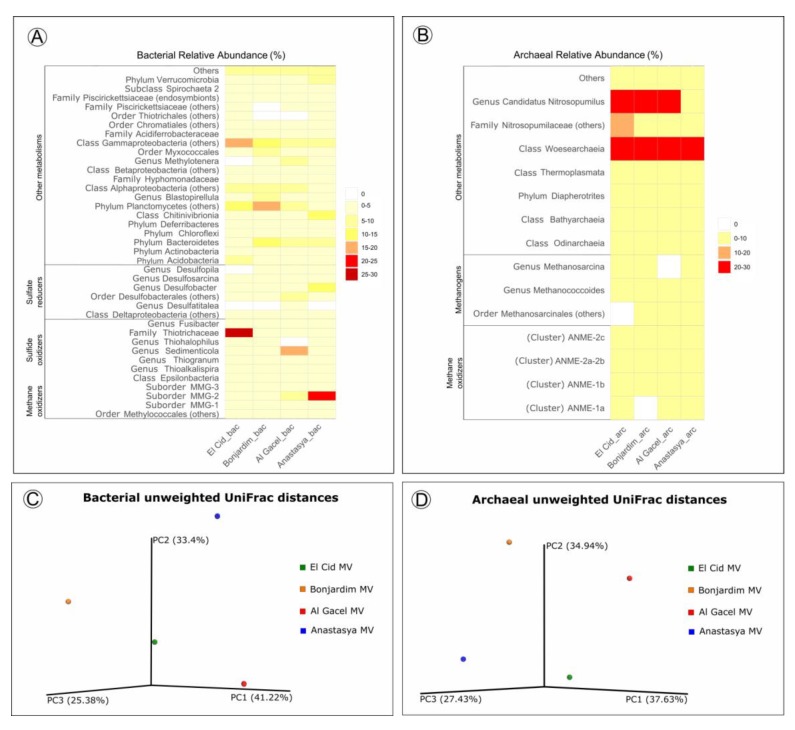
(**A,B**) Heatmap representing bacterial and archaeal relative abundances in each sample. (**C,D**) Beta-diversity measured with unweighted UniFrac metrics. PCoA plots are with data rarefied to the lowest number of sequences. PC1, PC2, and PC3, first, second, and third principal coordinate axes, respectively.

**Figure 8 microorganisms-08-00367-f008:**
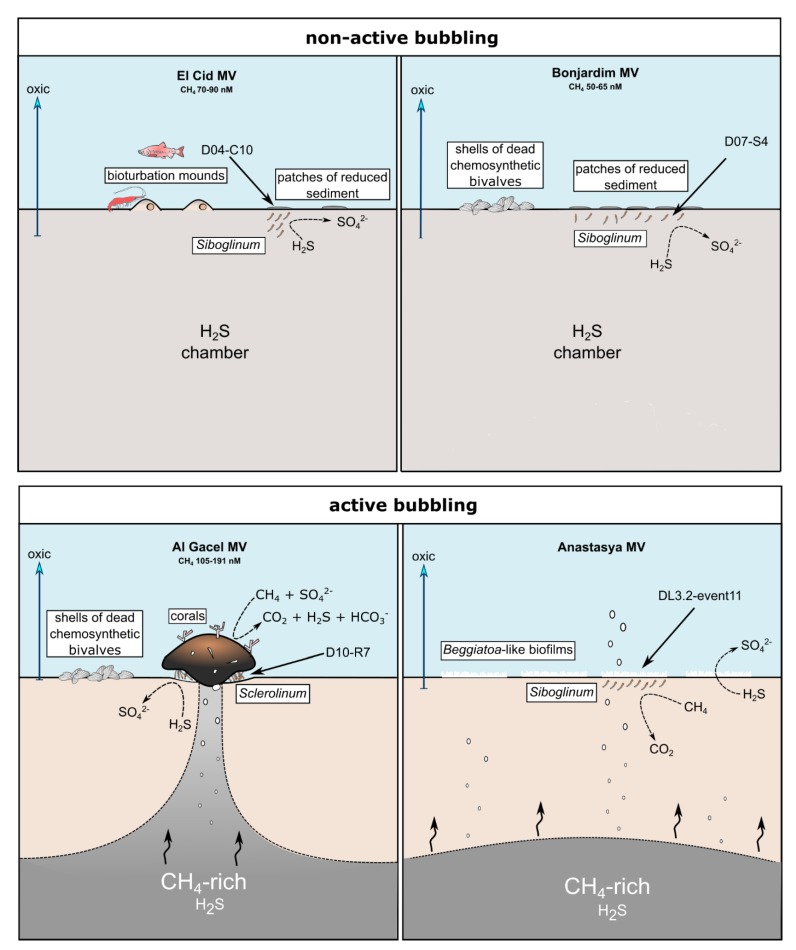
Scheme of the conditions given in the different sampling sites. Notice principal metabolism of siboglinids depending on seepage activity and presence or absence of other chemosynthetic organisms. Methane concentration values are given in Sánchez-Guillamón et al., 2015 [58].

**Table 1 microorganisms-08-00367-t001:** Exact sampling sites and variables’ measurement obtained from CH4 and CTD sensors of the ROVs.

Mud Volcano	Coordinates	Depth (m)	T (°C)	O_2_ (%)	CH_4_	pH	ORP (mV)	Description
El Cid	35° 26.32’ N 7° 29.03 W	1229	9.6	57	Yes	7.86	214	Grey mound surrounded by non-chemosynthetic fauna
Bonjardim	35° 27.52’ N 8° 59.99’ W	3051	2.8	6.14	Yes	7.91	188	Mud breccia with strong sulfidic smell and shells of chemosynthetic bivalves
Al Gacel	35° 26.47’ N 6° 58.27 W	791	10	54	Yes	7.88	149	Bottom of AOM authigenic carbonate from pockmark with active bubbling
Anastasya	36° 31.32’ N 7° 9.02 W	461			Yes			Black mud underneath white sulfur-oxidizing bacterial mat with active bubbling

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
