# Peer review of "Siboglinidae Tubes as an Additional Niche for Microbial Communities in the Gulf of Cádiz—A Microscopical Appraisal"

_microorganisms, 2020, doi:10.3390/microorganisms8030367_

Round 1

Reviewer 1 Report

The manuscript entitled “Siboglinidae tubes as an additional niche for microbial communities in the Gulf of Cádiz - a microscopical appraisal” by Rincon-Tomas et al. presents highly interesting and original data on preliminary characterization of the siboglinids microbiota recovered from four mud volcanoes in the Gulf of Cadiz. Employing light microscopy, transmission electron microscopy, scanning electron microscopy coupled to EDX and sequencing of bacterial and archaeal 16S rRNA genes authors characterized the tubes, tissues and microbiota of Siboglinum sp. worms, proposing the worms as a new microbial niche. Additionally, authors showed that seepage activities directly influence the composition of the microbial community colonized the studied worms. Despite the study does not discriminate between tube’s microbiota and worm‐tissue’s microbiota it still presents interesting set of data that might prompt the scientists to performed additional analysis to delineate the variation in the microbiota that depends on the presence or absence of the living worms.

Minor comments:

Lane 111: Probably authors meant “aqueous ethanol solution”

Lane 414: Correct to the “…degradation of the chitin…”

Author Response

Authors response to Reviewer nº 1
We are thankful for your constructive feedback and we have added your minor revisions:

Comment: Line 111: Probably authors meant “aqueous ethanol solution”
Reply: changed

Comment: Line 414: Correct to the “…degradation of the chitin…”
Reply: corrected

Reviewer 2 Report

Dear authors,

The submitted manuscript 686931 “Siboglinidae tubes as an additional niche for microbial communities in the Gulf of Cádiz — a microscopical appraisal” by Blanca Rincón-Tomás * , Francisco J. González , Luis Somoza , Kathrin Sauter , Pedro Madureira , Teresa Medialdea , Jens Carlsson , Michael Hoppert , Joachim Reitner describes the association of microbes to tubeworm of deep sea mud volacanoes in the Gulf of Cádiz. Authors used microscopy imaging (SEM, TEM) and 16S rRNA analysis for characterization of associated bacteria and archaea. The authors report that methanotrophic bacteria form biofilms on the worms. Mineralization and silicification of the tubes were observed. The authors conclude that tubeworms act as additional microbial niche in deep-sea ecosystems, where microbial biomass is concentrated.

The manuscript points to tubeworms as an important niche in deep-sea habitats for microbes. Likely, significant nutrient turnovers take place in those niches and crucially contribute to marine processes. Therefore, it is valuable to study those niches, however beyond the single organism concept. Tubeworms should be regarded as holobionts/metaorganisms with their associated microbiota. In particular, research has to be expanded from already known worm symbionts in the trophosome to the diverse associated microbiota. The manuscript promises the characterization of microbial communities, mainly by microscope imaging. Shown micrographs are impressive, but many conclusions made from those results are speculative correlations. For instance, the authors identified a tube-associated biofilm (dead!), which by definition seems to be a microbial layer. Based on the micrographs, I was not able to follow the conclusion that the biofilm consist of Methanotrophs, because no FISH probes were used for specific taxonomic localization. In addition, the sequencing approach or at least its description in the manuscript lacks much information, in the Material and Methods section as well as in the Results. No metrics on alpha- or beta diversity are shown and statistics are fully missing. For my understanding, those missing data are crucial to interpret and compare the imaging data. Specific comments are made below.

Specific comments to the authors in order of appearance in the manuscript:

Line 16: Gulf of Cádiz (El Cid MV, Bonjardim MV, Al Gacel MV, and Anastasya MV), are these certain locations? => can not be assumed as known here

Line 19: „chemosynthetic and non-chemosynthetic“ not necessary to mention

Line 24: EDX, please avoid abbreviations in the abstract

Line 43: better “live in symbiosis with …”

Line 46: Move sentence to the end of paragraph

Lines 61-70: Figure showing typical body plan of Siboglinids would be helpful

Line 72: I do not understand the mentioned implications

Line 78: 16S rRNA => correct throughout the text

Paragraph 2.1: How was the cross-contamination with ambient microbes from surrounding water avoided during sampling as well as ensured that sampling from depth does not damaged samples for microscopy? In which depths samples were taken? => Tab. 1 would be helpful in this section

Paragraph 2.2/3: What is the reason for the selection of certain samples for the different microscopy methods?

Paragraph 2.5: Why was a soil DNA extraction kit used for tubeworms? Are there already publications describing the choice of V3-V4 primers for tubeworms? More information on sequencing performance is necessary.

Paragraph 3.1: Would it be possible to do a genotyping of the tubeworms, for instance based on 18S rRNA, ITS etc. for exact taxonomic classification?

Tab. 1: I would recommend moving this table to the Methods section.

Fig. 3: Why were different filters used for visualization of fluorescence? Please describe differences of filters. Better, use wavelengths than filter names. How specific is the fluorescent dye? How can you conclude on a biofilm?

Micrographs: I am astonished by the few associated bacteria to worms. Can, beside the diversity, an abundance of bacteria/archaea on the worms estimated based on 16S data?

Paragraph 3.4: A biofilm of 1 – 2 µm makes me struggling. A biofilm should be a 3D-construct of bacterial cells embedded in an extracellular matrix. For me it sounds like one bacterial layer! I recommend avoiding “biofilm”. The “dead biofilm” might be a result of the sample preparation. How can you specify to methanotrophic bacteria only by imaging?

Paragraph 3.5: Please cite respective references supporting your statements, e.g. “typically found in water columns”.

Fig. 7: I recommend stating the taxonomic levels. Parts C and D of the figure do not add any important information. Nevertheless, what about alpha- and beta diversity?

Discussion: I recommend moving several sections to the Introduction part, e.g. conditions of sampling sites. Often, it is not clear, if the conclusion is based on the present study or preliminary studies. Mostly, the results do not necessarily point to the given explanations – it is often correlations and highly speculative.

Conclusion: As mentioned before, I do not see most of the statements verified by the results.

Round 2

Reviewer 2 Report

Dear authors,

after second revision and detailed point-to-point response, I have no more concers.